# Predict the Future from the Past? On the Temporal Data Distribution Shift in Financial Sentiment Classifications

## Yue Guo     Chenxi Hu     Yi Yang
The Hong Kong University of Science and Technology
yguoar@connect.ust.hk     chuak@connect.ust.hk     imyiyang@ust.hk

## Abstract

Temporal data distribution shift is prevalent in the financial text. How can a financial sentiment analysis system be trained in a volatile market environment that can accurately infer sentiment and be robust to temporal data distribution shifts? In this paper, we conduct an empirical study on the financial sentiment analysis system under temporal data distribution shifts using a real-world financial social media dataset that spans three years. We find that the fine-tuned models suffer from general performance degradation in the presence of temporal distribution shifts. Furthermore, motivated by the unique temporal nature of the financial text, we propose a novel method that combines out-of-distribution detection with time series modeling for temporal financial sentiment analysis. Experimental results show that the proposed method enhances the model's capability to adapt to evolving temporal shifts in a volatile financial market.

## 1 Introduction

Natural language processing (NLP) techniques have been widely adopted in financial applications, such as financial sentiment analysis, to facilitate investment decision-making and risk management (Loughran and McDonald, 2016; Kazemian et al., 2016; Bochkay et al., 2023). However, the *non-stationary* financial market environment can bring about significant changes in the data distribution between model development and deployment, which can degrade the model's performance over time and, consequently, its practical value. For example, a regime shift in the stock market refers to a significant change in the underlying economic or financial conditions. A regime shift, which may be triggered by changes in interest rates or political events, can significantly affect the market behavior and investor sentiment (Kritzman et al., 2012; Nystrup et al., 2018).

There has been limited research on the temporal dataset shift in the financial context. Existing NLP works on financial sentiment analysis follow the conventional approach that randomly splits a dataset into training and testing so that there is no distribution shift between training and testing (Malo et al., 2014; Cortis et al., 2017). However, in a real-world financial sentiment analysis system, there could be unpredictable distribution shifts between the data that is used to build the model (**in-sample data**) and the data that the model runs inference on (**out-of-sample data**). As a result, the practitioners often face a dilemma. If the model fits too well to the in-sample data, it may experience a drastic drop in the out-of-sample data if a distribution shift happens (such as a regime shift from a bull market to a bear market); if the model is built to minimize performance disruption, its performance may be unsatisfactory on the in-sample data as well as the out-of-sample data.

In this paper, we raise our first research question **RQ1**: *how does temporal data shift affect the robustness of financial sentiment analysis*? The question is not as trivial as it seems. For example, Guo et al. (2023a) find that language models are robust to temporal shifts in healthcare prediction tasks. However, financial markets may exhibit even more drastic changes. To answer this question, we systematically assess several language models, from BERT to GPT-3.5, with metrics that measure both the model capacity and robustness under temporal distribution shifts. In our monthly rolling-based empirical analysis, this dilemma between in-sample performance and out-of-sample performance is confirmed. We find that fine-tuning a pre-trained language model (such as BERT) fails to produce robust sentiment classification performance in the presence of temporal distribution shifts.

Moreover, we are interested in **RQ2**: *how to mitigate the performance degradation of financial sentiment analysis in the existence of temporal dis-*

*tribution shift*? Motivated by the unique temporal nature of financial text data, we propose a novel method that combines out-of-distribution (OOD) detection with autoregressive (AR) time series modeling. Experiments show that OOD detection can effectively identify the samples causing the model performance degradation (we refer to those samples as OOD data). Furthermore, the model performance on the OOD data is improved by an autoregressive time series modeling on the historical model predictions. As a result, the model performance degradation from in-sample data to out-of-sample data is alleviated.

This work makes two contributions to the literature. First, while sentiment analysis is a very well-studied problem in the financial context, the long-neglected problem is how to build a robust financial sentiment analysis model under the pervasive distribution shift. To our knowledge, this paper provides the first empirical evidence of the impact of temporal distribution shifts on financial sentiment analysis. Second, we propose a novel approach to mitigate the out-of-sample performance degradation while maintaining in-sample sentiment analysis utility. We hope this study contributes to the continuing efforts to build a more robust and accountable financial NLP system.

## 2 Temporal Distribution Shift in Financial Sentiment Analysis

In this section, we first define the task of financial sentiment analysis on temporal data. We then introduce two metrics for model evaluation under the data distribution shift.

### 2.1 Problem Formulation

The financial sentiment analysis model aims to classify a text input, such as a social media post or financial news, into positive or negative classes [1]. It can be expressed as a text classification model $M : M(X) \mapsto Y$. Conventionally, this task is modeled and evaluated on a non-temporal dataset, i.e., $(X, Y)$ consists of independent examples unrelated to each other in chronological order.

In the real world, financial text data usually exhibits temporal patterns corresponding to its occurrence time. To show this pattern, we denote $(X, Y) = \{(X_1, Y_1), ..., (X_N, Y_N)\}$, where

---

[1] We consider binary positive/negative prediction in this paper. Other financial analysis systems may have an additional *neutral* label (Huang et al., 2022).

$(X_t, Y_t)$ denotes a set of text and associated sentiment label collected from time $t$. Here $t$ could be at various time horizons, such as hourly, daily, monthly, or even longer horizon.

In the real-world scenarios, at time $t$, the sentiment classification model can only be trained with the data that is up to $t$, i.e., $\{(X_1, Y_1), ..., (X_t, Y_t))\}$. We denote the model trained with data up to period $t$ as $M_t$. In a continuous production system, the model is applied to the data $(X_{t+1}, Y_{t+1})$ in the next time period $t + 1$.

The non-stationary financial market environment leads to different data distributions at different periods, i.e., there is a temporal distribution shift. However, the non-stationary nature of the financial market makes it difficult to predict how data will be distributed in the next period. Without loss of generality, we assume $p(X_t, Y_t) \neq p(X_{t+1}, Y_{t+1})$ for any time $t$.

### 2.2 Evaluation Metrics

Unlike traditional financial sentiment analysis, temporal financial sentiment analysis trains the model on in-sample data and applies the model to the out-of-sample data. Therefore, in addition to the in-sample sentiment analysis performance, we also care about its generalization performance on out-of-sample data. In other words, we hope the model experiences minimal performance degradation even under significant temporal distribution shifts. Specifically, we use the standard classification metric $F1$-Score to measure the model performance. To measure the model generalization, we use $\Delta F1 = F1_{in} - F1_{out}$, where $F1_{in}$ and $F1_{out}$ are the $F1$-Score on in-sample and out-of-sample data respectively. An ideal financial sentiment analysis model would achieve high $F1_{in}$ and $F1_{out}$ and low $\Delta F1$ at the same time.

## 3 Experiment Setup

This section describes the evaluation setups on the dataset and models for temporal model analysis.

### 3.1 Dataset

We collect a time-stamped real-world financial text dataset from StockTwits[2], a Twitter-like social media platform for the financial and investing community. StockTwits data is also used in prior NLP work for financial sentiment analysis (Cortis et al., 2017), though the data is used in a conventional

---

[2] https://stocktwits.com/

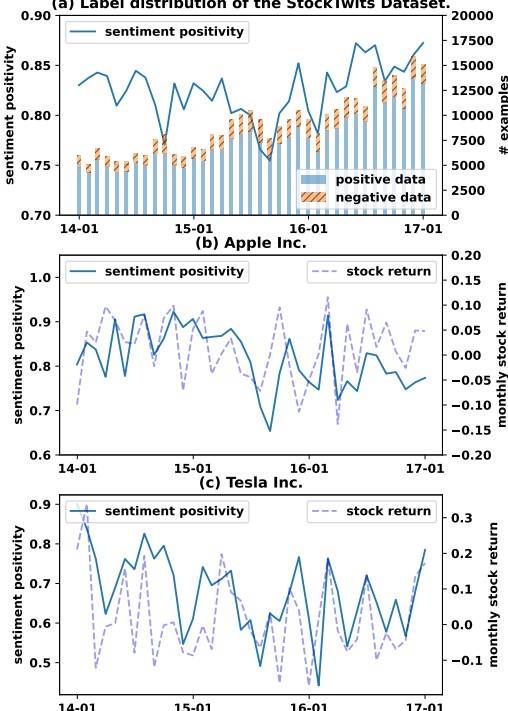

Figure 1: (a) A temporal view of the "sentiment positivity" (line) of the dataset and the number of examples with positive and negative labels (bar). (b) and (c): The "sentiment positivity" score (solid line) and the monthly stock price return (dashed line) of Apple Inc and Tesla Inc, respectively.

non-temporal setting. In our experiment, we collect all posts from StockTwits spanning from 2014-1-1 to 2016-12-31 [3]. We then filter the dataset by selecting those posts that contain a user-generated sentiment label bullish ("1") or bearish ("0"). The final dataset contains 418,893 messages, each associated with a publish date, providing helpful information for our temporal-based empirical analysis.

We provide some model-free evidence on the temporal distribution shift. First, we plot the "sentiment positivity" score for the monthly sentiment label distributions of the whole dataset in Figure 1 (a). The sentiment positivity is defined as the percentage of the positive samples, i.e., $\#pos/(\#pos + \#neg)$. It shows that while most messages are positive each month, the ratio between positive and negative samples fluctuates.

We then choose two representative companies, Apple Inc. and Tesla Inc., and filter the messages with the token "Apple" or "Tesla". We plot their sentiment positivity score in the solid lines in Figure 1 (b) and (c), respectively. We also plot the

monthly stock price return of Apple Inc. and Tesla Inc. in the dashed line in Figure 1 (b) and (c). It shows that the sentiment and the stock price movement are highly correlated, and the Spearman Correlation between the sentiment and the monthly return is 0.397 for Apple and 0.459 for Tesla. Moreover, the empirical conditional probability of labels given the specific token (i.e., "Apple", "Tesla") varies in different months. Taking together, we observe a temporal distribution shift in the financial social media text.

### 3.2 Sentiment Classification Models

We choose several standard text classification methods for sentiment classification, including (1) a simple logistic regression classifier that uses bag-of-words features of the input sentences, (2) an LSTM model with a linear classification layer on top of the LSTM hidden output, (3) three pretrained language models: BERT (base, uncased) (Devlin et al., 2019), RoBERTa (base) (Liu et al., 2019) and a finance domain specific pretrained model FinBERT (Yang et al., 2020); (4) the large language model GPT-3.5 (text-davinci-003) (Brown et al., 2020) with two-shot in-context learning.

## 4 Empirical Evaluation of Temporal Data Shift in Financial Sentiment Analysis

Our first research question aims to empirically examine if temporal data shift affects the robustness of financial sentiment analysis and to what extent. Prior literature has studied temporal distribution shifts in the healthcare domain and finds that pretrained language models are robust in the presence of temporal distribution shifts for healthcare-related prediction tasks such as hospital readmission (Guo et al., 2023a). However, financial markets and the financial text temporal shift are much more volatile. We empirically answer this research question using the experiment setup discussed in Section 3.

### 4.1 Training Strategy

Training a sentiment classification model on the time-series data is not trivial, as different utilization of the historical data and models would lead to different results in model performance and generalization. To comprehensively understand the model behavior under various settings, we summarize three training strategies by the different incorporation of the historical data and models.

---

[3]More recent year data is not easily accessible due to API restriction.

**Old Data, New Model** (ODNM): This training strategy uses all the available data up to time $t$, i.e.$\{(X_1, Y_1), ..., (X_t, Y_t)\}$ to train a new model $M_t$. With this training strategy, the sentiment analysis model is trained with the most diverse financial text data that is not restricted to the most recent period.

**New Data, New Model** (NDNM): For each time period $t$, a new model $M_t$ is trained with the latest data $(X_t, Y_t)$ collected in time $t$. This training strategy fits the model to the most recent data, which may have the most similar distribution to the out-of-sample data if there is no sudden change in the market environment.

**New Data, Old Model** (NDOM): Instead of training a new model from scratch every time, we update the model trained at the previous time with the latest data. Specifically, in time $t$, the parameters of the model $M_t$ are initialized with the parameters from $M_{t-1}$ and continuously learn from $(X_t, Y_t)$. This training strategy inherits the knowledge from past data but still adapts to more recent data.

For GPT-3.5, we use two-shot in-context learning to prompt the model. The in-context examples are randomly selected from $(X_t, Y_t)$. The prompt is "Perform financial sentiment classification: text:{a positive example} label:positive; text:{a negative example} label:negative; text:{testing example} label:".

## 4.2 Rolling-based Empirical Test

To empirically examine temporal sentiment classification models, we take a rolling-based approach. We divide the dataset by month based on the timestamp of each text. Since we have three years of StockTwits data, we obtain 36 monthly subsets. For each month $t$, we train a sentiment classification model $M_t$ (Section 3.2, except GPT-3.5) using a training strategy in Section 4.1 that uses data up to month $t$. For evaluation, we consider the testing samples from $(X_t, Y_t)$ as the in-sample and the testing samples from $(X_{t+1}, Y_{t+1})$ as out-of-sample. This rolling-based approach simulates a real-world continuous production setup. Since we have 36 monthly datasets, our temporal-based empirical analysis is evaluated on $36-1 = 35$ monthly datasets (except the last month). We report the average performance in $F1$-score and $\Delta F1$ as 2.2. The train/validate/test split is by 7:1.5:1.5 randomly.

## 4.3 Empirical Results

We evaluate different sentiment classification models using different training strategies [4]. We present the main empirical results on the original imbalanced dataset in Table 1, averaged over the 35 monthly results. An experiment using the balanced dataset after up-sampling the minor examples is presented in Appendix A, from which similar conclusions can be drawn. To better understand the model performance over time, we plot the results by month in Figure 2, using BERT and NDOM strategy as an example. The monthly results of NDNM and ODNM on BERT are shown in Appendix B. Furthermore, to compare the performance and the robustness against data distribution shift among the training strategies, we plot the in-sample performance $F1_{in}(avg)$ in Figure 3, and the performance drop $\Delta F1(avg)$ in Figure 4 by the training strategies. The $F1_{out}(avg)$ is supplemented in Appendix B.

We have the following observations from the experimental results: **The performance drop in the out-of-sample prediction is prevalent, especially for the negative label.** All $\Delta F1$ of the fine-tuned models in table 1 are positive, indicating that all fine-tuned models suffer from performance degradation when serving the models to the next month's data. Such performance drop is especially significant for the negative label, indicating that the minority label suffers even more in the model generalization. The results remain after up-sampling the minority label, as shown in Appendix A.

**NDOM training strategy achieves the best in-sample performance**, yet fails to generalize on out-of-sample. Table 1 and Figure 3 show that NDOM has the highest $F1$-score among the three training strategies, especially for the in-sample prediction. **Training with ODNM strategy is most robust against data distribution shift.** As shown in Table 1 and Figure 4, among the three training strategies, ODNM has the smallest performance drop in out-of-sample predictions. It suggests that using a long range of historical data can even out the possible distribution shift in the dataset and improve the model's robustness.

**There is a trade-off between in-sample performance and out-of-sample performance.** As stated above, the NDOM strategy achieves the best

---

[4]Since LR and LSTM are trained using dataset-specific vocabulary, the NDOM training strategy, which uses the old model's vocabulary, is not applicable.

| | | $F1_{in}(pos)$ ↑ | $F1_{out}(pos)$ ↑ | $\Delta F1(pos)$ ↓ | $F1_{in}(neg)$ ↑ | $F1_{out}(neg)$ ↑ | $\Delta F1(neg)$ ↓ | $F1_{in}(avg)$ ↑ | $F1_{out}(avg)$ ↑ | $\Delta F1(avg)$ ↓ |
|---|---|---|---|---|---|---|---|---|---|---|
| LogisticRegression | ODNM | 89.9 | 89.7 | **0.27** | 34.8 | 32.7 | **2.10** | 62.3 | 61.2 | **1.18** |
| | NDNM | **91.2** | **90.1** | 1.07 | **46.2** | **36.1** | 10.04 | **68.7** | **63.1** | 5.55 |
| LSTM | ODNM | **91.4** | **90.6** | **0.73** | **55.8** | **51.3** | **4.51** | **73.6** | **71.0** | **2.62** |
| | NDNM | 90.4 | 89.1 | 1.32 | 44.6 | 35.2 | 9.42 | 67.5 | 62.2 | 5.37 |
| BERT | ODNM | 89.9 | 89.5 | **0.43** | 44.9 | 43.0 | **1.95** | 67.4 | 66.2 | **1.19** |
| | NDNM | 91.5 | 90.4 | 1.13 | 53.5 | 45.2 | 8.27 | 72.5 | 67.8 | 4.70 |
| | NDOM | **93.1** | **92.1** | 0.95 | **65.1** | **59.5** | 5.67 | **79.1** | **75.8** | 3.31 |
| RoBERTa | ODNM | 91.0 | 90.8 | **0.24** | 48.8 | 47.1 | **1.35** | 69.9 | 68.9 | **0.97** |
| | NDNM | 91.7 | 90.6 | 1.12 | 56.8 | 50.0 | 6.80 | 74.3 | 70.3 | 3.96 |
| | NDOM | **93.3** | **92.6** | 0.72 | **67.4** | **63.2** | 4.18 | **80.4** | **77.9** | 2.45 |
| FinBERT | ODNM | 89.4 | 89.2 | **0.25** | 40.3 | 38.5 | **1.76** | 64.9 | 63.9 | **1.00** |
| | NDNM | 91.2 | 90.0 | 1.24 | 50.7 | 41.2 | 9.52 | 71.0 | 65.6 | 5.38 |
| | NDOM | **92.6** | **91.7** | 0.83 | **60.9** | **54.9** | 6.05 | **76.7** | **73.3** | 3.44 |
| GPT-3.5 | | 81.8 | 82.5 | -0.70 | 51.0 | 52.5 | -1.50 | 66.4 | 67.5 | -1.10 |

Table 1: Performance of different models under New Data New Model (NDNM), New Data Old Model (NDOM), and Old Data New Model(ODNM) training strategies. The numbers are averaged over the monthly evaluation over three years.

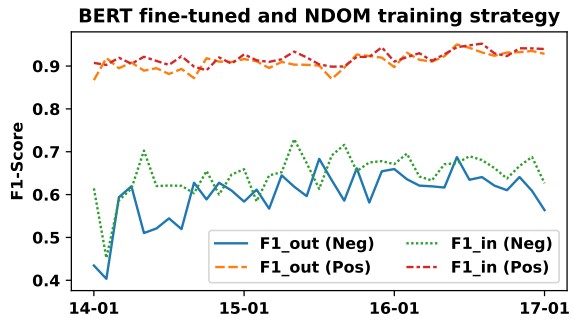

Figure 2: BERT's in-sample and out-of-sample prediction using the NDOM strategy. The performance gaps between the in-sample and out-of-sample prediction are significant, especially for the negative label.

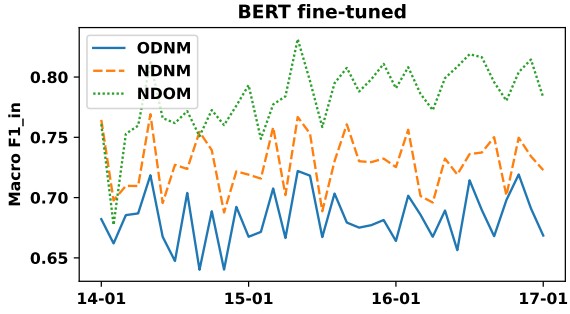

Figure 3: The performance comparison $F1_{in}(avg)$ using three training strategies in BERT.

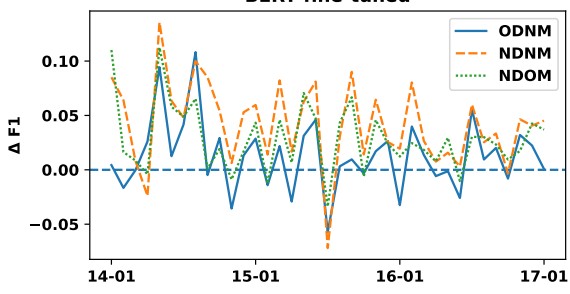

Figure 4: The performance drop $\Delta F1(avg)$ using three training strategies in BERT.

in-sample performance but suffers significant out-of-sample degradation. ODNM, conversely, has the slightest performance degradation, yet its overall prediction capability is limited. From a practical perspective, both strategies are not ideal. First, accurately identifying the market sentiments is essential for a financial sentiment classification model to build a portfolio construction. Second, it is also important that financial sentiment classification models produce stable prediction performance so that the subsequent trading portfolio, driven by the prediction outcomes, can have the least disruption.

**In GPT-3.5, the performance drop is not observed, but the in-sample and out-of-sample performance falls behind the fine-tuned models.** It indicates that increasing the model size and training corpus may reduce the performance drop when facing a potential distribution shift. However, GPT-3.5 is not ideal, as its performance on the in-sample and out-of-sample data is significantly worse than the fine-tuned models, which aligns with the findings from (Guo et al., 2023b). Moreover, as the training corpus in GPT-3.5 covers the data from 2014 to 2017, our test set may not be genuinely out-of-distribution regarding GPT-3.5, which may also result in alleviating the performance drop.

### 4.4 Additional Analysis

We conduct additional analysis to understand the problem of performance degradation further. We examine the relationship between distribution shift and performance drop. To measure the distribution shift between in-sample data $(X_t, Y_t)$ in month $t$ and out-of-sample data $(X_{t+1}, Y_{t+1})$ in month $t+1$, we follow the steps: 1) For each token $v$ in the vocabulary $\mathcal{V}$,[5] we estimate the empirical probability

---
[5] $\mathcal{V}$ is the vocabulary from the pretrained tokenizer.

| | Spearmanr ($p$-value) |
|---|---|
| BERT | 0.385 (0.020) |
| RoBERTa | 0.345 (0.039) |
| FinBERT | 0.281 (0.096) |

Table 2: Spearman correlations between the performance drop ($\Delta F1$) and in-sample to out-of-sample distribution shift.

| Important Features |
|---|
| devices, incbr, incbry, nakd, incnn, **bears**, pharmaceuticals, **patience**, incara, ptx, **bulls**, **release**, incsla, paper, **fall**, **dump**, pump, **strong**, **advanced**, incphs, ride, imnp, added, caterpillar, kellogg, **shorts**, plx, owens, **dilution**, **squeeze** |

Table 3: Top 30 important words identified by the Logistic Regression model. The bold words are the financial sentiment words, and the unbolded words are the spurious correlated words. The model assigns high importance to the many spurious words and is prone to be influenced by spurious correlations.

$p_t(y|v), p_{t+1}(y|v), p_t(v), p_{t+1}(v)$. 2) We measure the distribution shift from $(X_t, Y_t)$ to $(X_{t+1}, Y_{t+1})$ as the weighted sum of the KL-divergence (KLD) between the two conditional probability distributions: $\sum_v p_t(v) KLD(p_t(y|v), p_{t+1}(y|v))$.

We then compute the Spearman correlation between the performance drop $\Delta F1$ and the distribution shift from in-sample data $(X_t, Y_t)$ to out-of-sample data $(X_{t+1}, Y_{t+1})$. The result is shown in Table 2, showing a significant positive correlation between performance drop and distribution shift. Therefore, a more significant distribution shift, primarily caused by financial market turbulence, can lead to more severe model performance degradation. This problem is especially problematic because the performance degradation may exacerbate sentiment-based trading strategy during a volatile market environment, leading to significant investment losses.

Second, we provide empirical evidence that the models are prone to be influenced by spurious correlations. Generally, a sentiment classification model makes predictions on the conditional probability $p(y|v)$ based on some sentiment words $v$. Ideally, such predictions are effective if there is no distribution shift and the model can successfully capture the sentiment words (e.g., $v =$ bearish, bullish, and so on). However, if a model is affected by spurious correlations, it undesirably associates the words with no sentiment with the label. The model generalization will be affected when the correlations between the spurious words and

the sentiment label change in the volatile financial market. For example, suppose the model makes predictions based on the spurious correlated words $p(y|$"Tesla"$)$. When the market sentiment regarding Tesla fluctuates, the model's robustness will be affected.

We use the most explainable logistic regression model as an example to provide evidence for spurious correlations. The logistic regression model assigns a coefficient to each word in the vocabulary, suggesting the importance of the word contributed to the prediction. We fit a logistic regression model to our dataset and then extract the top 30 words with the highest coefficients (absolute value). The extracted words are listed in Table 3, with bold words indicating the sentiment words and the unbolded words as the spurious words. We can see that most words the model regards as important are not directly connected to the sentiments. As a result, the performance of model prediction $p(y|v)$ is likely influenced by the changing financial sentiment in the volatile markets.

# 5  Mitigating Model Degradation under Temporal Data Shift

In the previous section, our analysis reveals a consistent performance degradation in financial sentiment classification models on out-of-sample data. In this section, we explore possible ways to mitigate the degradation. As our previous analysis shows that the performance degradation is correlated with the distribution shift, it is reasonable to infer that the performance degradation is caused by the failure of the out-of-distribution (OOD) examples. This observation motivates us to propose a two-stage method to improve the model robustness under temporal data distribution shifts. Firstly, we train an OOD sample detector to detect whether an upcoming sample is out of distribution. If not, we still use the model trained on in-sample data on this sample. If yes, we propose using an autoregressive model from time series analysis to simulate the model prediction towards the future.

## 5.1  Mitigation Method

This subsection introduces the mitigation method for temporal financial sentiment analysis.

### 5.1.1  Detecting OOD Samples

As the model trained on the historical data experiences performance degradation on the future data under distribution shift, we first employ an OOD

detection mechanism to determine the ineffective future data. To train the OOD detector, we collect a new dataset that contains the in-distribution (ID) data (label=0) and OOD data (label=1) regarding a model $M_t$. The labeling of the OOD dataset is based on whether $M_t$ can correctly classify the sample, given the condition that the in-sample classifier can correctly classify the sample.

Specifically, let $i,j$ denote the indexes of time satisfying $i < j$, given a target sentiment classifier $M_i$, and a sample $(x_j, y_j)$ which can be correctly classified by the in-sample sentiment model $M_j$, the OOD dataset assigns the label by the rule

$$OOD(M_i, x_j) = \begin{cases} 0, & \text{if } M_i(x_j) = y_j \\ 1, & M_i(x_j) \neq y_j \end{cases} \quad (1)$$

After collecting the OOD dataset, we train an OOD classifier $f(M, x)$ on the OOD dataset to detect whether a sample $x$ is OOD concerning the model $M$. The classifier is a two-layer multi-layer perceptron (MLP) on the [CLS] token of $M(x)$, i.e.,

$$f(M, x) = W_2(GELU(W_1(M^{[CLS]}(x))+b_1))+b_2 \quad (2)$$

The classifier is optimized through the cross-entropy loss on the OOD dataset. During training, the sentiment model $M$ parameters are fixed, and only the parameters of the MLP (i.e., $W_1, W_2, b_1, b_2$) are updated by gradient descent. The parameters of the OOD classifier are used across all sentiment models $M \in \{M_1, ..., M_N\}$ regardless of time.

During inference, given a sentiment model $M_t$ and an out-of-sample $x_{t+1}$ , we compute $f(M_t, x_{t+1})$ to detect whether $x_{t+1}$ is OOD regarding to $M_t$. If $x_{t+1}$ is not an OOD sample, we infer the sentiment of $x_{t+1}$ by $M_t$. Otherwise, $x_{t+1}$ is regarded as an OOD sample, and $M_t$ may suffer from model degradation. Therefore, we predict the sentiment of $x_{t+1}$ by the autoregressive model from time-series analysis to avoid potential ineffectiveness.

### 5.1.2 Autoregressive Modeling

In time series analysis, an autoregressive (AR) model assumes the future variable can be expressed by a linear combination of its previous values and on a stochastic term. Motivated by this, as the distribution in the future is difficult to estimate directly, we assume the prediction from a future model can also be expressed by the combination of the past models' predictions. Specifically, given an OOD sample $x_{t+1}$ detected by the OOD classifier, the prediction $\hat{y}_{t+1}$ is given by linear regression on the predictions from the past models $M_t, ..., M_{t-p+1}$, i.e.,

$$\hat{y}_{t+1} = \sum_{k=0}^{p-1} \alpha_k M_{t-k}(x_{t+1}) + \epsilon \quad (3)$$

, where $\alpha_k$ is the regression coefficient and $\epsilon$ is the error term estimated from the past data. Moreover, $p$ is the order of the autoregressive model determined empirically.

For temporal data in financial sentiment classification, the future distribution is influenced by an aggregation of recent distributions and a stochastic term. Using an AR model on previous models' predictions can capture this feature. The AR modeling differs from a weighted ensemble method that assigns each model a fixed weight. In our method, weights assigned to past models are determined by how recently they were trained or used, with more recent models receiving higher weights.

### 5.2 Experiment Setup

To train the OOD detector and estimate the parameters in the AR model, we use data from 2014-01 to 2015-06. We split the data each month by 7:1.5:1.5 for sentiment model training, detector/AR model training, and model testing, respectively. The data from 2015-07 to 2016-12 is used to evaluate the effectiveness of the mitigation method.

To train the OOD detector, we use AdamW optimizer and grid search for the learning rate in $[2\times10^{-3}, 2\times10^{-4}, 2\times10^{-5}]$, batch size in $[32, 64]$. When building the OOD dataset regarding $M_t$, we use the data that happened within three months starting from $t$.

To estimate the parameters in the AR model, for a training sample $(x_t, y_t)$, we collect the predictions of $x_t$ from $M_{t-1}, .., M_{t-p}$, and train a regression model to predict $y_t$, for $t$ from 2014-01+$p$ to 2015-06. We empirically set the order of the AR model $p$ as 3 in the experiment.

### 5.3 Baselines

Existing NLP work has examined model robustness on out-of-domain data in a non-temporal shift setting. We experiment with two popular methods, spurious tokens masking (Wang et al., 2022) and counterfactual data augmentation (Wang and Culotta, 2021), to examine their capability in miti-

| BERT | | | |
|---|---|---|---|
| | Precision | Recall | F1 |
| ID | 0.99 | 0.8 | 0.88 |
| OOD | 0.23 | 0.86 | 0.37 |
| Accuracy | | | 0.8 |
| FinBERT | | | |
| | Precision | Recall | F1 |
| ID | 1 | 0.81 | 0.9 |
| OOD | 0.15 | 0.91 | 0.26 |
| Accuracy | | | 0.82 |

Table 4: The performance of the OOD detector of BERT and FinBERT. The most crucial indicator is the recall of OOD data, as we want to retrieve as much OOD data as possible.

| | $F1_{in}(avg)\uparrow$ | $F1_{out}(avg)\uparrow$ | $\Delta F1(avg)\downarrow$ |
|---|---|---|---|
| BERT | 80.13 | 78.07 | 2.05 |
| +STM | 78.04 | 75.87 | 2.18 |
| +CDA | 76.91 | 74.88 | 2.02 |
| +Ours | 80.13 | **78.70** | **1.42** |
| RoBERTa | 81.87 | 80.25 | 1.62 |
| +STM | 81.12 | 79.14 | 1.98 |
| +CDA | 79.68 | 77.59 | 2.09 |
| +Ours | 81.87 | **80.61** | **1.26** |
| FinBERT | 77.46 | 75.44 | 2.01 |
| +STM | 76.57 | 74.49 | 2.08 |
| +CDA | 73.92 | 72.02 | 1.91 |
| +Ours | 77.46 | **76.09** | **1.36** |

Table 5: The mitigation results. Our method improves the performance of the sentiment model on out-of-sample data and reduces the performance drop.

gating the performance drop under temporal data distribution shift.

**Spurious Tokens Masking (Wang et al., 2022) (STM)** is motivated to improve the model robustness by reducing the spurious correlations between some tokens and labels. STM identifies the spurious tokens by conducting cross-dataset stability analysis. While genuine and spurious tokens have high importance, "spurious" tokens tend to be important for one dataset but fail to generalize to others. Therefore, We identify the "spurious tokens" as those with high volatility in the attention score across different months from 2014-01 to 2015-06. Then, we mask the identified spurious tokens during training and inference on the data from 2015-07 to 2016-12.

**Counterfactual Data Augmentation (Wang and Culotta, 2021) (CDA)** improves the model robustness by reinforcing the impact of the causal clues. It first identifies the causal words by a matching algorithm and then generates the counterfactual data by replacing the identified causal word with its antonym. Like STM, we identify causal words on the monthly datasets from 2014-01 to 2015-06.

More details of the setup of the two baseline methods are presented in Appendix C.

### 5.4 Mitigation Results

First, we analyze the performance of the OOD detector. Table 4 shows the classification reports of the OOD detector of BERT and FinBERT on the test set. The recall of OOD data is the most crucial indicator of the model performance, as we want to retrieve as much OOD data as possible before applying it to the AR model to avoid potential model degradation. As shown in table 4, the detector can achieve the recall of 0.86 and 0.91 for OOD data in BERT and FinBERT, respectively, indicating the adequate capability to identify the data that the sentiment models will wrongly predict. As the dataset is highly unbalanced towards the ID data, the relatively low precision in OOD data is expected. Nevertheless, the detector can achieve an accuracy of around 0.8 on the OOD dataset.

Table 5 shows the results of the mitigation methods under the NDOM training strategy. We only apply our mitigation method to the out-of-sample prediction. For BERT, RoBERTa, and FinBERT, our method reduces the performance drop by 31%, 26%, and 32%, respectively. Our results show that The AR model can improve the model performance on OOD data. As a result, the overall out-of-sample performance is improved, and the model degradation is alleviated.

Another advantage of our methods is that, unlike the baseline methods, our method does not require re-training the sentiment models. Both baseline methods re-train the sentiment models on the newly generated datasets, either by data augmentation or spurious tokens masking, at the cost of influencing the model performance. Our proposed methods avoid re-training the sentiment models and improve the out-of-sample prediction with aggregation on the past models.

## 6 Related Works

**Temporal Distribution Shift.** While temporal distribution shift has been studied in other contexts such as healthcare (Guo et al., 2022), there is no systematic empirical study of temporal distribution shift in the finance domain. Moreover, although a prior study in the healthcare domain has shown

that the large language models can significantly mitigate temporal distribution shifts on healthcare-related tasks such as readmission prediction (Guo et al., 2023a), financial markets are more volatile, so is the financial text temporal shift. Our empirical study finds that fine-tuned models suffer from model degradation under temporal distribution shifts.

**Financial Sentiment Analysis.** NLP techniques have gained widespread adoption in the finance domain (Loughran and McDonald, 2016; Yang et al., 2020; Bochkay et al., 2023; Shah et al., 2022; Wu et al., 2023; Chuang and Yang, 2022). One of the essential applications is financial sentiment classification, where the inferred sentiment is used to guide trading strategies and financial risk management (Kazemian et al., 2016). However, prior NLP work on financial sentiment classification has not explored the temporal distribution shift problem, a common phenomenon in financial text. This work aims to investigate the financial temporal distribution shift empirically and proposes a mitigation method.

## 7   Conclusions

In this paper, we empirically study the problem of distribution shift over time and its adverse impacts on financial sentiment classification models. We find a consistent yet significant performance degradation when applying a sentiment classification model trained using in-sample (past) data to the out-of-sample (future) data. The degradation is driven by the data distribution shift, which, unfortunately, is the nature of dynamic financial markets. To improve the model's robustness against the ubiquitous distribution shift over time, we propose a novel method that combines out-of-distribution detection with autoregressive (AR) time series modeling. Our method is effective in alleviating the out-of-sample performance drop.

Given the importance of NLP in real-world financial applications and investment decision-making, there is an urgent need to understand the weaknesses, safety, and robustness of NLP systems. We raise awareness of this problem in the context of financial sentiment classification and conduct a temporal-based empirical analysis from a practical perspective. The awareness of the problem can help practitioners improve the robustness and accountability of their financial NLP systems and also calls for developing effective NLP systems that are robust to temporal data distribution shifts.

## Limitations

This paper has several limitations to improve in future research. First, our temporal analysis is based on the monthly time horizon, so we only analyze the performance degradation between the model built in the current month $t$ and the following month $t + 1$. Future analysis can investigate other time interval granularity, such as weekly or annual. Second, our data is collected from a social media platform. The data distribution on social media platforms may differ from other financial data sources, such as financial news articles or analyst reports. Thus, the robustness of language models on those types of financial text data needs to be examined, and the effectiveness of our proposed method on other types of financial text data warrants attention. Future research can follow our analysis pipeline to explore the impact of data distribution shifts on financial news or analyst reports textual analysis. Third, our analysis focuses on sentiment classification performance degradation. How performance degradation translates into economic losses is yet to be explored. Trading simulation can be implemented in a future study to understand the economic impact of the problem better.

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
