# OpenReview forum: "Predict the Future from the Past? On the Temporal Data Distribution Shift in Financial Sentiment Classifications"
_EMNLP/2023/Conference — EMNLP 2023 Main_

### Official Review · Reviewer_EjqB · 2023-07-31

**Soundness:** 3

**Excitement:**

3: Ambivalent: It has merits (e.g., it reports state-of-the-art results, the idea is nice), but there are key weaknesses (e.g., it describes incremental work), and it can significantly benefit from another round of revision. However, I won't object to accepting it if my co-reviewers champion it.

**Paper Topic And Main Contributions:**

The paper investigates temporal data distribution shifts in financial sentiment analysis. It identifies performance degradation in fine-tuned models under such shifts. To address this, a novel method combining out-of-distribution detection with time series modeling is proposed. Experimental results demonstrate its effectiveness in adapting to evolving temporal shifts in a volatile financial market. Overall, the study offers valuable insights and a promising solution to improve sentiment analysis in dynamic financial environments.

**Reasons To Accept:**

1.	The paper is well structured.
2.	The paper addresses a significant problem in the field of financial sentiment analysis. Temporal data distribution shifts are prevalent in financial text, and developing robust models to handle such shifts is crucial for accurate sentiment analysis in volatile markets.
3.	The authors conduct the study using a real-world financial social media dataset spanning three years. This approach ensures the findings are grounded in real data, making the results more applicable and reliable.

**Reasons To Reject:**

1.	I understand that sentiment will change with financial trends, but the carrier of sentiment features is the semantics expressed in words. Why does the author think that the semantic features corresponding to sentiment at different times are significantly different?
2.	This work seems like a stacking and permutation of various models, lacking innovation in models.
3.	Since the authors mention the volatility of financial sentiment data, they should emphasize how to design models or specific fine-tune methods to address this issue.

**Reproducibility:**

3: Could reproduce the results with some difficulty. The settings of parameters are underspecified or subjectively determined; the training/evaluation data are not widely available.

**Reviewer Confidence:**

3: Pretty sure, but there's a chance I missed something. Although I have a good feel for this area in general, I did not carefully check the paper's details, e.g., the math, experimental design, or novelty.

---

> ### Author Rebuttal · Authors · 2023-08-29
>
> Dear Reviewer EjqB:
>
> Thank you very much for your thoughtful and positive review of our work. We truly appreciate the time you took to evaluate our work on analyzing and addressing the temporal data distribution shift problem in financial sentiment classifications. Your insights and feedback are invaluable to us.
>
> We would like to make some clarifications on the reviews:
>
> Regarding reason 1 to reject, we agree that semantic words convey the sentiment. In other words, the model makes predictions on the conditional probability $p(y|v)$ based on some sentiment words $v$. Ideally, such predictions are effective if there is no distribution shift and the model can successfully capture the sentiment words (e.g., $v=$ bearish, bullish, etc.). However, we also find that the models are prone to be influenced by spurious correlations. For example, if the model makes predictions based on the spurious correlated words $p(y|\text{"Microsoft"})$ when the market sentiment regarding Microsoft fluctuates, the model robustness will also be affected.
> We use the most explainable Logistic Regression model as an example to provide evidence for such spurious correlations. After fitting the model to our dataset, we extract the top 30 words with the highest coefficients (absolute value), which are listed in the table below:
>
> |                                                                                                                                        Important Features                                                                                                                                        |
> |:------------------------------------------------------------------------------------------------------------------------------------------------------------------------------------------------------------------------------------------------------------------------------------------------:|
> |     devices, incbr, incbry, nakd, incnn, **bears**, pharmaceuticals, **patience**, incara, ptx, **bulls**, **release**, incsla, paper, **fall**, **dump**, pump, **strong**, **advanced**, incphs, ride, imnp, added, caterpillar, kellogg, **shorts**, plx, owens, **dilution**, **squeeze**    |
>
> We can see that most of the features the model regards as important are not directly connected to the sentiments except the bolded words that convey financial sentiments. As a result, the model prediction $p(y|v)$ is likely to be influenced by the changing financial sentiment in the volatile markets. This experiment further confirms that the findings of our work are non-trivial and counter-intuitive, raising awareness and addressing the importance of the negative impact of temporal distribution shifts on financial sentiment analysis.
>
> Regarding reason 2 to reject, we would like to explain that our work is not simply stacking and permutation of various models. Our contributions are twofold: (1) a novel OOD detection method: facing the challenge of the unknown and unpredictable distribution shifts, we propose a new method to conduct OOD detection by training a classifier on the OOD dataset generated on the historical data. The OOD detection is effective and avoids the requirement to know the future distribution. (2) A method to improve model performance. After identifying the OOD data, as the in-domain model is expected to be ineffective in the OOD data, we propose a method to improve model performance with the autoregressive model from the time-series analysis. The intuition of using a time-series analysis model is that the main objective of the time-series analysis is to predict the future value from the past values; therefore, it is very suitable for our scenario to improve the model’s future performance. Experiments prove that our method is effective.
>
> Regarding reason 3 to reject, we agree that designing specific fine-tuning methods could be a solution to improve the model's robustness. However, the empirical study in section 4.3 shows a trade-off between model absolute performance (in-sample performance) and model robustness to distribution shift (out-of-sample performance). As a result, it is not easy for a fine-tuning method to improve in-sample and out-of-sample performance simultaneously (see the results of the baseline models). Considering these difficulties, we do not use a new model or a fine-tuning method for our task. Instead, we propose to predict the future label with a time-series analysis method. To our knowledge, we are the first to propose to apply the time-series analysis method to improve model robustness on NLP tasks. Experiments show that our method is effective in both the in-sample and out-of-sample performance.
>
> If you have any further questions or suggestions for collaboration, please feel free to reach out. We are always open to fruitful discussions and potential partnerships.
>
> Once again, thank you for your support and feedback. We look forward to sharing more of our work with you in the future.
>
>
> Best regards,
> Authors

---

### Official Review · Reviewer_mkMy · 2023-08-02

**Soundness:** 4

**Excitement:**

4: Strong: This paper deepens the understanding of some phenomenon or lowers the barriers to an existing research direction.

**Paper Topic And Main Contributions:**

Topic:

This paper empirically studies the financial sentiment analysis system under temporal data distribution shifts.
And this paper proposes a novel method that combines out-of-distribution detection with time series modeling for temporal financial sentiment analysis.

Contributions:

This paper builds a robust financial sentiment analysis model under the pervasive distribution shift.

This paper provides the first empirical evidence of the impact of temporal distribution shifts on financial sentiment analysis.

This paper proposes a novel approach to mitigate the out-of-sample performance degradation while maintaining in-sample sentiment analysis utility.


**Questions For The Authors:**

Question 1: What is the calculation formula of alpha_k (regression coefficient)? It refers to “weights are determined by how recently they were trained”

**Reasons To Accept:**

This paper proposes a novel method that combines out-of-distribution (OOD) detection with autoregressive (AR) time series modeling to alleviate model performance degradation.

The analysis of Temporal Data Shift in Financial Sentiment (Section 4) is valuable.

**Reasons To Reject:**

N/A

**Reproducibility:**

4: Could mostly reproduce the results, but there may be some variation because of sample variance or minor variations in their interpretation of the protocol or method.

**Reviewer Confidence:**

3: Pretty sure, but there's a chance I missed something. Although I have a good feel for this area in general, I did not carefully check the paper's details, e.g., the math, experimental design, or novelty.

---

> ### Author Rebuttal · Authors · 2023-08-29
>
> Dear Reviewer mkMy:
>
> Thank you very much for your thoughtful and positive review of our work. We truly appreciate the time you took to evaluate our work on analyzing and addressing the temporal data distribution shift problem in financial sentiment classifications. Your insights and feedback are invaluable to us.
>
> For question 1, we use the data from 2014-01 to 2015-06 to train the mitigation models, and then use the data from 2015-07 to 2016-12 to test the method's effectiveness (the same setting for all baselines). Rolling from 2014-01+p to 2015-06 (p is the order of the autoregressive model), for each example $x_t$ on time stamp $t$, we collect the past model predictions $M_{t-p}(x_t), M_{t-p+1}(x_t), …, M_{t-1}(x_t)$, and the ground-truth label $y_t$. We train a linear regression model to predict $y_t$ based on past model predictions. $\alpha_k$ is the regression coefficient learned by the linear regression model. Empirically, we find that the well-trained regression model assigns a higher weight to the model closer to the time $t$. We will explain this experiment setting more clearly in the revised version of the paper.
>
> If you have any further questions or suggestions for collaboration, please feel free to reach out. We are always open to fruitful discussions and potential partnerships.
>
> Once again, thank you for your support and positive feedback. We look forward to sharing more of our work with you in the future.
>
> Best regards,
>
> Authors

---

### Official Review · Reviewer_MD99 · 2023-08-05

**Soundness:** 3

**Excitement:**

4: Strong: This paper deepens the understanding of some phenomenon or lowers the barriers to an existing research direction.

**Paper Topic And Main Contributions:**

The paper explores financial sentiment analysis in the presence of temporal data distribution shifts using a real-world financial social media dataset. The main contributions include:

- Conducting an empirical study of the performance of fine-tuned models in a volatile financial market with temporal data distribution shifts.
- Proposing a method that combines out-of-distribution detection with time-series modeling to improve the adaptability of financial sentiment analysis systems in the face of evolving temporal shifts.


**Reasons To Accept:**

The paper tackles the timely and challenging problem of applying sentiment analysis to financial text in the presence of temporal data distribution shifts. The empirical study and proposed method contribute valuable insights into improving the performance of financial sentiment analysis models in volatile market environments.

**Reasons To Reject:**

No significant concerns or issues were identified.

**Reproducibility:**

2: Would be hard pressed to reproduce the results. The contribution depends on data that are simply not available outside the author's institution or consortium; not enough details are provided.

**Reviewer Confidence:**

2: Willing to defend my evaluation, but it is fairly likely that I missed some details, didn't understand some central points, or can't be sure about the novelty of the work.

---

> ### Author Rebuttal · Authors · 2023-08-29
>
> Dear Reviewer MD99:
>
> Thank you very much for your thoughtful and positive review of our work. We truly appreciate the time you took to evaluate our work on analyzing and addressing the temporal data distribution shift problem in financial sentiment classifications. Your insights and feedback are invaluable to us.
>
> We are delighted that you recognize our empirical study of the model degradation phenomenon in a volatile financial market is timely and practical, and the proposed method is effective in this environment with temporal data distribution shifts. Your recognition of our efforts in this area means a lot to our team.
>
> If you have any further questions or suggestions for collaboration, please feel free to reach out. We are always open to fruitful discussions and potential partnerships.
>
> Once again, thank you for your support and positive feedback. We look forward to sharing more of our work with you in the future.
>
> Best regards,
>
> Authors

---

### Meta-Review · Area_Chair_f4b3 · 2023-09-17

**Recommendation:** 5

**Metareview:**

The reviewers are in general agreement that this paper could be accepted in EMNLP-2023. The reviewers also raised some concerns about this submission. The authors have provided a rebuttal that appeared to alleviate some concerns. A suggestion was made to accept the paper as Main Conference. The reviewers' concerns are also suggested to be addressed in the final version.

---

### Decision · Program_Chairs · 2023-10-07

**Decision:**

Accept-Main

**Comment:**

The reviewers are in general agreement that this paper could be accepted in EMNLP-2023. The reviewers also raised some concerns about this submission. The authors have provided a rebuttal that appeared to alleviate some concerns. A suggestion was made to accept the paper as Main Conference. The reviewers' concerns are also suggested to be addressed in the final version.